# Domain Generalization with MixStyle

**Kaiyang Zhou[1], Yongxin Yang[1], Yu Qiao[2], Tao Xiang[1]**
[1]University of Surrey, UK
[2]Shenzhen Institutes of Advanced Technology, Chinese Academy of Sciences, Shenzhen, China
`k.zhou.vision@gmail.com`
`{yongxin.yang, t.xiang}@surrey.ac.uk`
`yu.qiao@siat.ac.cn`

## Abstract

Though convolutional neural networks (CNNs) have demonstrated remarkable ability in learning discriminative features, they often generalize poorly to unseen domains. Domain generalization aims to address this problem by learning from a set of source domains a model that is generalizable to any unseen domain. In this paper, a novel approach is proposed based on probabilistically mixing instance-level feature statistics of training samples across source domains. Our method, termed MixStyle, is motivated by the observation that visual domain is closely related to image style (e.g., photo vs. sketch images). Such style information is captured by the bottom layers of a CNN where our proposed style-mixing takes place. Mixing styles of training instances results in novel domains being synthesized implicitly, which increase the domain diversity of the source domains, and hence the generalizability of the trained model. MixStyle fits into mini-batch training perfectly and is extremely easy to implement. The effectiveness of MixStyle is demonstrated on a wide range of tasks including category classification, instance retrieval and reinforcement learning.

## 1 Introduction

Key to automated understanding of digital images is to compute a compact and informative feature representation. Deep convolutional neural networks (CNNs) have demonstrated remarkable ability in representation learning, proven to be effective in many visual recognition tasks, such as classifying photo images into 1,000 categories from ImageNet (Krizhevsky et al., 2012) and playing Atari games with reinforcement learning (Mnih et al., 2013). However, it has long been discovered that the success of CNNs heavily relies on the i.i.d. assumption, i.e. training and test data should be drawn from the same distribution; when such an assumption is violated even just slightly, as in most real-world application scenarios, severe performance degradation is expected (Hendrycks & Dietterich, 2019; Recht et al., 2019).

Domain generalization (DG) aims to address such a problem (Zhou et al., 2021; Blanchard et al., 2011; Muandet et al., 2013; Li et al., 2018a; Zhou et al., 2020b; Balaji et al., 2018; Dou et al., 2019; Carlucci et al., 2019). In particular, assuming that multiple source domains containing the same visual classes are available for model training, the goal of DG is to learn models that are robust against data distribution changes across domains, known as domain shift, so that the trained model can generalize well to any unseen domains. Compared to the closely related and more widely studied domain adaptation (DA) problem, DG is much harder in that no target domain data is available for the model to analyze the distribution shift in order to overcome the negative effects. Instead, a DG model must rely on the source domains and focus on learning domain-invariant feature representation in the hope that it would remain discriminative given target domain data.

A straightforward solution to DG is to expose a model with a large variety of source domains. Specifically, the task of learning domain-invariant and thus generalizable feature representation becomes easier when data from more diverse source domains are available for the model. This would reduce the burden on designing special models or learning algorithms for DG. Indeed, model training with large-scale data of diverse domains is behind the success of existing commercial face recognition or vision-based autonomous driving systems. A recent work by Xu et al. (2021) also emphasizes the

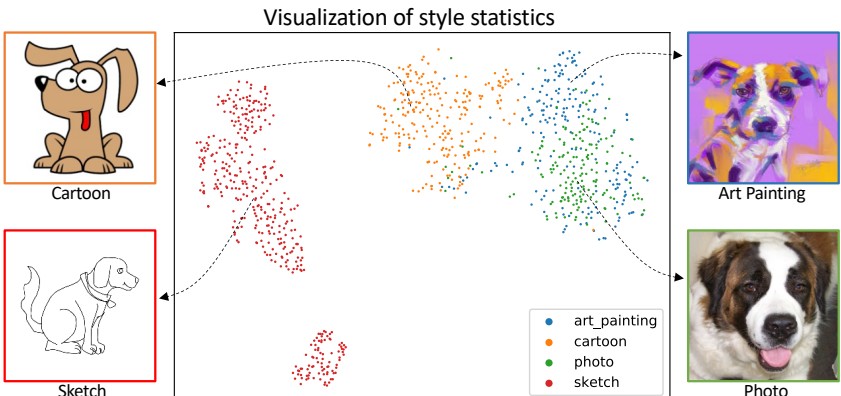

Figure 1: 2-D t-SNE (Maaten & Hinton, 2008) visualization of the style statistics (concatenation of mean and standard deviation) computed from the first residual block's feature maps of a ResNet-18 (He et al., 2016) trained on four distinct domains (Li et al., 2017). It is clear that different domains are well separated.

importance of diverse training distributions for out-of-distribution generalization. However, collecting data of a large variety of domains is often costly or even impossible. It thus cannot be a general solution to DG.

In this paper, a novel approach is proposed based on probabilistically mixing instance-level feature statistics of training samples across source domains. Our model, termed *MixStyle*, is motivated by the observation that visual domain is closely related to image style. An example is shown in Fig. 1: the four images from four different domains depict the same semantic concept, i.e. dog, but with distinctive styles (e.g., characteristics in color and texture). When these images are fed into a deep CNN, which maps the raw pixel values into category labels, such style information is removed at the output. However, recent style transfer studies (Huang & Belongie, 2017; Dumoulin et al., 2017) suggest that such style information is preserved at the bottom layers of the CNN through the instance-level feature statistics, as shown clearly in Fig. 4. Importantly, since replacing such statistics would lead to replaced style while preserving the semantic content of the image, it is reasonable to assume that mixing styles from images of different domains would result in images of (mixed) new styles. That is, more diverse domains/styles can be made available for training a more domain-generalizable model.

Concretely, our MixStyle randomly selects two instances of different domains and adopts a probabilistic convex combination between instance-level feature statistics of bottom CNN layers. In contrast to style transfer work (Huang & Belongie, 2017; Dumoulin et al., 2017), no explicit image synthesis is necessary meaning much simpler model design. Moreover, MixStyle perfectly fits into modern mini-batch training. Overall, it is very easy to implement with only few lines of code. To evaluate the effectiveness as well as the general applicability of MixStyle, we conduct extensive experiments on a wide spectrum of datasets covering category classification (Sec. 3.1), instance retrieval (Sec. 3.2), and reinforcement learning (Sec. 3.3). The results demonstrate that MixStyle can significantly improve CNNs' cross-domain generalization performance.[1]

## 2 METHODOLOGY

### 2.1 BACKGROUND

Normalizing feature tensors with instance-specific mean and standard deviation has been found effective for removing image style in style transfer models (Ulyanov et al., 2016; Huang & Belongie, 2017; Dumoulin et al., 2017). Such an operation is widely known as instance normalization (IN, Ulyanov et al. (2016)). Let $x \in \mathbb{R}^{B \times C \times H \times W}$ be a batch of tensors, with $B$, $C$, $H$ and $W$

---

[1]Source code can be found at https://github.com/KaiyangZhou/mixstyle-release.

denoting the dimension of batch, channel, height and width, respectively, IN is formulated as

$$\text{IN}(x) = \gamma \frac{x - \mu(x)}{\sigma(x)} + \beta, \tag{1}$$

where $\gamma, \beta \in \mathbb{R}^C$ are learnable affine transformation parameters, and $\mu(x), \sigma(x) \in \mathbb{R}^{B \times C}$ are mean and standard deviation computed across the spatial dimension within each channel of each instance (tensor), i.e.

$$\mu(x)_{b,c} = \frac{1}{HW} \sum_{h=1}^{H} \sum_{w=1}^{W} x_{b,c,h,w}, \tag{2}$$

and

$$\sigma(x)_{b,c} = \sqrt{\frac{1}{HW} \sum_{h=1}^{H} \sum_{w=1}^{W} (x_{b,c,h,w} - \mu(x)_{b,c})^2}. \tag{3}$$

Huang & Belongie (2017) introduced adaptive instance normalization (AdaIN), which simply replaces the scale and shift parameters in Eq. (1) with the feature statistics of style input $y$ to achieve arbitrary style transfer:

$$\text{AdaIN}(x) = \sigma(y) \frac{x - \mu(x)}{\sigma(x)} + \mu(y). \tag{4}$$

## 2.2 MIXSTYLE

Our method, MixStyle, draws inspiration from AdaIN. However, rather than attaching a decoder for image generation, MixStyle is designed for the purpose of regularizing CNN training by perturbing the style information of source domain training instances. It can be implemented as a plug-and-play module inserted between CNN layers of, e.g., a supervised CNN classifier, without the need to explicitly generate an image of new style.

More specifically, MixStyle mixes the feature statistics of two instances with a random convex weight to simulate new styles. In terms of implementation, MixStyle can be easily integrated into mini-batch training. Given an input batch $x$, MixStyle first generates a reference batch $\tilde{x}$ from $x$. When domain labels are given, $x$ is sampled from two different domains $i$ and $j$, e.g., $x = [x^i, x^j]$ ($x^i$ and $x^j$ have the same batch size). Then, $\tilde{x}$ is obtained by swapping the position of $x^i$ and $x^j$, followed by a shuffling operation along the batch dimension applied to each batch, i.e. $\tilde{x} = [\text{Shuffle}(x^j), \text{Shuffle}(x^i)]$. See Fig. 2(a) for an illustration. In cases where domain labels are unknown, $x$ is randomly sampled from the training data, and $\tilde{x}$ is simply obtained by $\tilde{x} = \text{Shuffle}(x)$ (see Fig. 2(b)). Fig. 4 shows that sub-domains exist within each domain, so even if two instances of the same domain are sampled, new domain could be synthesized. After shuffling, MixStyle computes the mixed feature statistics by

$x = [\ x_1\ \ x_2\ \ x_3\ \ x_4\ \ x_5\ \ x_6\ ]$

$\tilde{x} = [\ x_5\ \ x_6\ \ x_4\ \ x_3\ \ x_1\ \ x_2\ ]$

(a) Shuffling batch w/ domain label

$x = [\ x_1\ \ x_2\ \ x_3\ \ x_4\ \ x_5\ \ x_6\ ]$

$\tilde{x} = [\ x_6\ \ x_1\ \ x_5\ \ x_3\ \ x_2\ \ x_4\ ]$

(b) Shuffling batch w/ random shuffle

Figure 2: A graphical illustration of how a reference batch is generated. Domain label is denoted by color.

$$\gamma_{mix} = \lambda \sigma(x) + (1 - \lambda)\sigma(\tilde{x}), \tag{5}$$
$$\beta_{mix} = \lambda \mu(x) + (1 - \lambda)\mu(\tilde{x}), \tag{6}$$

where $\lambda \in \mathbb{R}^B$ are instance-wise weights sampled from the Beta distribution, $\lambda \sim Beta(\alpha, \alpha)$ with $\alpha \in (0, \infty)$ being a hyper-parameter. Unless specified otherwise, we set $\alpha$ to 0.1 throughout this paper. Finally, the mixed feature statistics are applied to the style-normalized $x$,

$$\text{MixStyle}(x) = \gamma_{mix} \frac{x - \mu(x)}{\sigma(x)} + \beta_{mix}. \tag{7}$$

In practice, we use a probability of 0.5 to decide if MixStyle is activated or not in the forward pass. At test time, no MixStyle is applied. Note that gradients are blocked in the computational graph of $\mu(\cdot)$ and $\sigma(\cdot)$. MixStyle can be implemented with only few lines of code. See Algorithm 1 in Appendix A.1 for the PyTorch-like pseudo-code.

Table 1: Leave-one-domain-out generalization results on PACS.

| Method | Art | Cartoon | Photo | Sketch | Avg |
|---|---|---|---|---|---|
| MMD-AAE | 75.2 | 72.7 | 96.0 | 64.2 | 77.0 |
| CCSA | 80.5 | 76.9 | 93.6 | 66.8 | 79.4 |
| JiGen | 79.4 | 75.3 | 96.0 | 71.6 | 80.5 |
| CrossGrad | 79.8 | 76.8 | 96.0 | 70.2 | 80.7 |
| Epi-FCR | 82.1 | 77.0 | 93.9 | 73.0 | 81.5 |
| Metareg | 83.7 | 77.2 | 95.5 | 70.3 | 81.7 |
| L2A-OT | 83.3 | 78.2 | 96.2 | 73.6 | 82.8 |
| ResNet-18 | 77.0±0.6 | 75.9±0.6 | 96.0±0.1 | 69.2±0.6 | 79.5 |
| + Manifold Mixup | 75.6±0.7 | 70.1±0.9 | 93.5±0.7 | 65.4±0.6 | 76.2 |
| + Cutout | 74.9±0.4 | 74.9±0.6 | 95.9±0.3 | 67.7±0.9 | 78.3 |
| + CutMix | 74.6±0.7 | 71.8±0.6 | 95.6±0.4 | 65.3±0.8 | 76.8 |
| + Mixup (w/o label interpolation) | 74.7±1.0 | 72.3±0.9 | 93.0±0.4 | 69.2±0.2 | 77.3 |
| + Mixup | 76.8±0.7 | 74.9±0.7 | 95.8±0.3 | 66.6±0.7 | 78.5 |
| + DropBlock | 76.4±0.7 | 75.4±0.7 | 95.9±0.3 | 69.0±0.3 | 79.2 |
| + MixStyle w/ random shuffle | 82.3±0.2 | **79.0**±0.3 | **96.3**±0.3 | 73.8±0.9 | 82.8 |
| + MixStyle w/ domain label | **84.1**±0.4 | 78.8±0.4 | 96.1±0.3 | **75.9**±0.9 | **83.7** |

## 3 EXPERIMENTS

### 3.1 GENERALIZATION IN CATEGORY CLASSIFICATION

**Dataset and implementation details.** We choose the PACS dataset (Li et al., 2017), a commonly used domain generalization (DG) benchmark concerned with domain shift in image classification. PACS consists of four domains, i.e. Art Painting, Cartoon, Photo and Sketch, with totally 9,991 images of 7 classes. As shown in Fig. 1, the domain shift mainly corresponds to image style changes. For evaluation, a model is trained on three domains and tested on the remaining one. Following prior work (Li et al., 2019; Zhou et al., 2020a), we use ResNet-18 (He et al., 2016) as the classifier where MixStyle is inserted after the 1st, 2nd and 3rd residual blocks. Our code is based on Dassl.pytorch (Zhou et al., 2020c).[2]

**Baselines.** Our main baselines are general-purpose regularization methods including Mixup (Zhang et al., 2018b), Manifold Mixup (Verma et al., 2019), DropBlock (Ghiasi et al., 2018), CutMix (Yun et al., 2019) and Cutout (DeVries & Taylor, 2017), which are trained using the same training parameters as MixStyle and the optimal hyper-parameter setup as reported in their papers. We also compare with the existing DG methods which reported state-of-the-art performance on PACS. These include domain alignment-based CCSA (Motiian et al., 2017) and MMD-AAE (Li et al., 2018b), Jigsaw puzzle-based JiGen (Carlucci et al., 2019), adversarial gradient-based CrossGrad (Shankar et al., 2018), meta-learning-based Metareg (Balaji et al., 2018) and Epi-FCR (Li et al., 2019), and data augmentation-based L2A-OT (Zhou et al., 2020a).

**Comparison with general-purpose regularization methods.** The results are shown in Table 1. Overall, we observe that the general-purpose regularization methods do not offer any clear advantage over the vanilla ResNet-18 in this DG task, while MixStyle improves upon the vanilla ResNet-18 with a significant margin. Compared with Mixup, MixStyle is 5.2% better on average. Recall that Mixup also interpolates the output space, we further compare with a variant of Mixup in order to demonstrate the advantage of mixing style statistics at the feature level over mixing images at the pixel level for DG—following Sohn et al. (2020), we remove the label interpolation in Mixup and sample the mixing weights from a uniform distribution of [0, 1]. Still, MixStyle outperforms this new baseline with a large margin, which justifies our claim. MixStyle and DropBlock share some commonalities in that they are both applied to feature maps at multiple layers, but MixStyle significantly outperforms DropBlock in all test domains. The reason why DropBlock is ineffective here is because dropping out activations mainly encourages a network to mine discriminative patterns, but does not reinforce the ability to cope with unseen styles, which is exactly what MixStyle aims to achieve: by synthesizing "new" styles (domains) MixStyle regularizes the network to become more

---

[2]https://github.com/KaiyangZhou/Dassl.pytorch.

robust to domain shift. In addition, it is interesting to see that on Cartoon and Photo, MixStyle w/ random shuffle obtains slightly better results. The reason might be because there exist sub-domains in a source domain (see Fig. 4(a-c)), which allow random shuffling to produce more diverse "new" domains that lead to a more domain-generalizable model.

**Comparison with state-of-the-art DG methods.** Overall, MixStyle outperforms most DG methods by a clear margin, despite being a much simpler method. The performance of MixStyle w/ domain label is nearly 1% better on average than the recently introduced L2A-OT. From a data augmentation perspective, MixStyle and L2A-OT share a similar goal—to synthesize data from pseudo-novel domains. MixStyle accomplishes this goal through mixing style statistics at the feature level. Whereas L2A-OT works at the pixel level: it trains an image generator by maximizing the domain difference (measured by optimal transport) between the original and the generated images, which introduces much heavier computational overhead than MixStyle in terms of GPU memory and training time. It is worth noting that MixStyle's domain label-free version is highly competitive: its 82.8% accuracy is on par with L2A-OT's.

## 3.2 Generalization in Instance Retrieval

**Dataset and implementation details.** We evaluate MixStyle on the person re-identification (re-ID) problem, which aims to match people across disjoint camera views. As each camera view is itself a distinct domain, person re-ID is essentially a cross-domain image matching problem. Instead of using the standard protocol where training and test data come from the same camera views, we adopt the *cross-dataset* setting so test camera views are never seen during training. Specifically, we train a model on one dataset and then test its performance on the other dataset. Two commonly used re-ID datasets are adopted: Market1501 (Zheng et al., 2015) and Duke (Ristani et al., 2016; Zheng et al., 2017). Ranking accuracy and mean average precision (mAP) are used as the performance measures (displayed in percentage). We test MixStyle on two CNN architectures: ResNet-50 (He et al., 2016) and OSNet (Zhou et al., 2019). The latter was designed specifically for re-ID. In both architectures, MixStyle is inserted after the 1st and 2nd residual blocks. Our code is based on Torchreid (Zhou & Xiang, 2019).[3]

**Baselines.** We compare with three baseline methods: 1) The vanilla model, which serves as a strong baseline; 2) DropBlock, which was the top-performing competitor in Table 1; 3) RandomErase (Zhong et al., 2020), a widely used regularization method in the re-ID literature (similar to Cutout).

**Results.** The results are reported in Table 2. It is clear that only MixStyle consistently outperforms the strong vanilla model under both settings with considerable margins, while DropBlock and RandomErase are unable to show any benefit. Notably, RandomErase, which simulates occlusion by erasing pixels in random rectangular regions with random values, has been used as a default trick when training re-ID CNNs. However, RandomErase shows a detrimental effect in the cross-dataset re-ID setting. Indeed, similar to DropBlock, randomly erasing pixels offers no guarantee to improve the robustness when it comes to domain shift.

## 3.3 Generalization in Reinforcement Learning

Though RL has been greatly advanced by using CNNs for feature learning in raw pixels (Mnih et al., 2013), it has been widely acknowledged that RL agents often overfit training environments while generalize poorly to unseen environments (Cobbe et al., 2019; Igl et al., 2019).

**Dataset and implementation details.** We conduct experiments on Coinrun (Cobbe et al., 2019), a recently introduced RL benchmark for evaluating the generalization performance of RL agents. As shown in Fig. 3(a), the goal in Coinrun is to control a character to collect golden coins while avoiding both stationary and non-stationary obstacles. We follow Igl et al. (2019) to construct and train our RL agent: the CNN architecture used in IMPALA (Espeholt et al., 2018) is adopted as the policy network, and is trained by the Proximal Policy Optimization (PPO) algorithm (Schulman et al., 2017). Please refer to Igl et al. (2019) for further implementation details. MixStyle is inserted after the 1st and 2nd convolutional sequences. Training data are sampled from 500 levels while test

---

[3]https://github.com/KaiyangZhou/deep-person-reid.

Table 2: Generalization results on the cross-dataset person re-ID task.

| Model | Market1501→Duke | | | | Duke→Market1501 | | | |
|---|---|---|---|---|---|---|---|---|
| | mAP | R1 | R5 | R10 | mAP | R1 | R5 | R10 |
| ResNet-50 | 19.3 | 35.4 | 50.3 | 56.4 | 20.4 | 45.2 | 63.6 | 70.9 |
| + RandomErase | 14.3 | 27.8 | 42.6 | 49.1 | 16.1 | 38.5 | 56.8 | 64.5 |
| + DropBlock | 18.2 | 33.2 | 49.1 | 56.3 | 19.7 | 45.3 | 62.1 | 69.1 |
| + MixStyle w/ random shuffle | **23.8** | 42.2 | 58.8 | **64.8** | 24.1 | 51.5 | 69.4 | 76.2 |
| + MixStyle w/ domain label | 23.4 | **43.3** | **58.9** | 64.7 | **24.7** | **53.0** | **70.9** | **77.8** |
| OSNet | 25.9 | 44.7 | 59.6 | 65.4 | 24.0 | 52.2 | 67.5 | 74.7 |
| + RandomErase | 20.5 | 36.2 | 52.3 | 59.3 | 22.4 | 49.1 | 66.1 | 73.0 |
| + DropBlock | 23.1 | 41.5 | 56.5 | 62.5 | 21.7 | 48.2 | 65.4 | 71.3 |
| + MixStyle w/ random shuffle | 27.2 | **48.2** | **62.7** | **68.4** | 27.8 | 58.1 | 74.0 | **81.0** |
| + MixStyle w/ domain label | **27.3** | 47.5 | 62.0 | 67.1 | **29.0** | **58.2** | **74.9** | 80.9 |

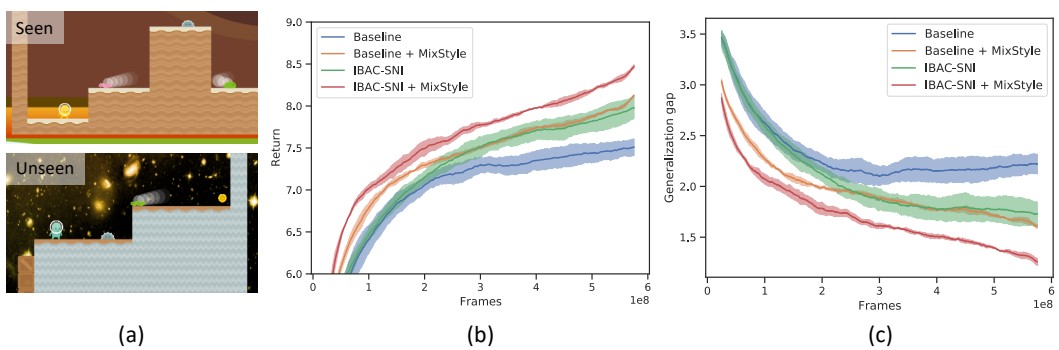

(a)            (b)            (c)

Figure 3: (a) Coinrun benchmark. (b) Test performance in unseen environments. (c) Difference between training and test performance.

data are drawn from new levels of only the highest difficulty. As domain labels are difficult to define, we use the random shuffle version of MixStyle. Our code is built on top of Igl et al. (2019).[4]

**Baselines.** Following Igl et al. (2019), we train strong baseline models and add MixStyle on top of them to see whether MixStyle can bring further improvements. To this end, we train two baseline models: 1) Baseline, which combines weight decay and data augmentation;[5] 2) IBAC-SNI (the $\lambda = 0.5$ version), the best-performing model in Igl et al. (2019) which is based on selective noise injection.

**Results.** The test performance is shown in Fig. 3(b). Comparing Baseline (blue) with Baseline+MixStyle (orange), we can see that MixStyle brings a significant improvement. Interestingly, the variance is also significantly reduced by using MixStyle, as indicated by the smaller shaded areas (for both orange and red lines). These results strongly demonstrate the effectiveness of MixStyle in enhancing generalization for RL agents. When it comes to the stronger baseline IBAC-SNI (green), MixStyle (red) is able to bring further performance gain, suggesting that MixStyle is complementary to IBAC-SNI. This result also shows the potential of MixStyle as a plug-and-play component to be combined with other advanced RL methods. It is worth noting that Baseline+MixStyle itself is already highly competitive with IBAC-SNI. Fig. 3(c) shows the generalization gap from which it can been seen that the models trained with MixStyle (orange & red) clearly generalize faster and better than those without using MixStyle (blue & green).

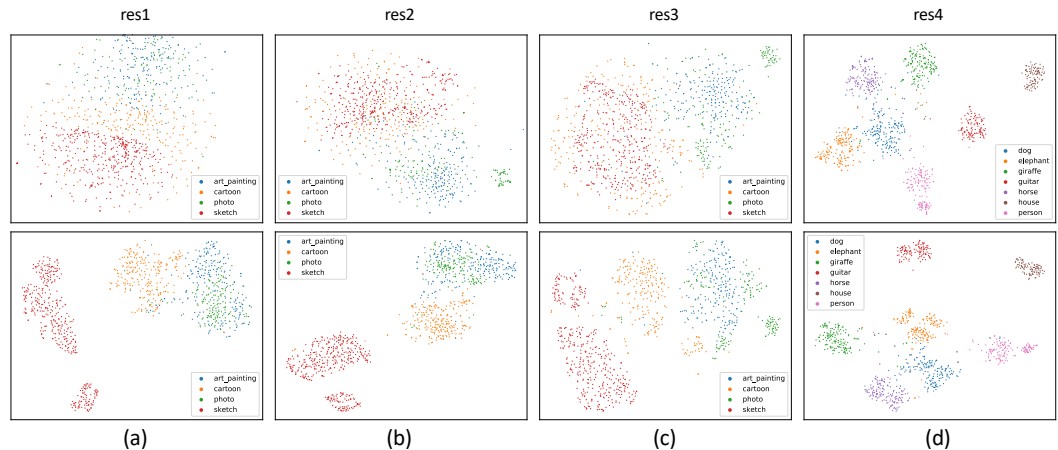

Figure 4: 2-D visualization of flattened feature maps (top) and the corresponding style statistics (bottom). `res1-4` denote the four residual blocks in order in a ResNet architecture. We observe that `res1` to `res3` contain domain-related information while `res4` encodes label-related information.

Table 3: Ablation study on where to apply MixStyle in the ResNet architecture.

(a) Category classification on PACS.

| Model | Accuracy |
|---|---|
| ResNet-18 | 79.5 |
| + MixStyle (`res1`) | 80.1 |
| + MixStyle (`res12`) | 81.6 |
| + MixStyle (`res123`) | **82.8** |
| + MixStyle (`res1234`) | 75.6 |
| + MixStyle (`res14`) | 76.3 |
| + MixStyle (`res23`) | 81.7 |

(b) Cross-dataset person re-ID.

| Model | mAP |
|---|---|
| ResNet-50 | 19.3 |
| + MixStyle (`res1`) | 22.6 |
| + MixStyle (`res12`) | **23.8** |
| + MixStyle (`res123`) | 22.0 |
| + MixStyle (`res1234`) | 10.2 |
| + MixStyle (`res14`) | 11.1 |
| + MixStyle (`res23`) | 20.6 |

## 3.4 ANALYSIS

**Where to apply MixStyle?** We repeat the experiments on PACS (category classification) and the re-ID datasets (instance retrieval) using the ResNet architecture. Given that a standard ResNet model has four residual blocks denoted by `res1-4`, we train different models with MixStyle applied to different layers. For notation, `res1` means MixStyle is applied after the first residual block; `res12` means MixStyle is applied after both the first and second residual blocks; and so forth. The results are shown in Table 3. We have the following observations. 1) Applying MixStyle to multiple lower-level layers generally achieves a better performance—for instance, `res12` is better than `res1` on both tasks. 2) Different tasks favor different combinations—`res123` achieves the best performance on PACS, while on the re-ID datasets `res12` is the best. 3) On both tasks, the performance plunges when applying MixStyle to the last residual block. This makes sense because `res4` is the closest to the prediction layer and tends to capture semantic content (i.e. label-sensitive) information rather than style. In particular, `res4` is followed by an average-pooling layer, which essentially forwards the mean vector to the prediction layer and thus forces the mean vector to capture label-related information. As a consequence, mixing the statistics at `res4` breaks the inherent label space. This is clearer in Fig. 4: the features and style statistics in `res1-3` exhibit clustering patterns based on domains while those in `res4` have a high correlation with class labels.

---

[4] `https://github.com/microsoft/IBAC-SNI`.

[5] We do not use batch normalization (Ioffe & Szegedy, 2015) or dropout (Srivastava et al., 2014) because they are detrimental to the performance, as shown by Igl et al. (2019).

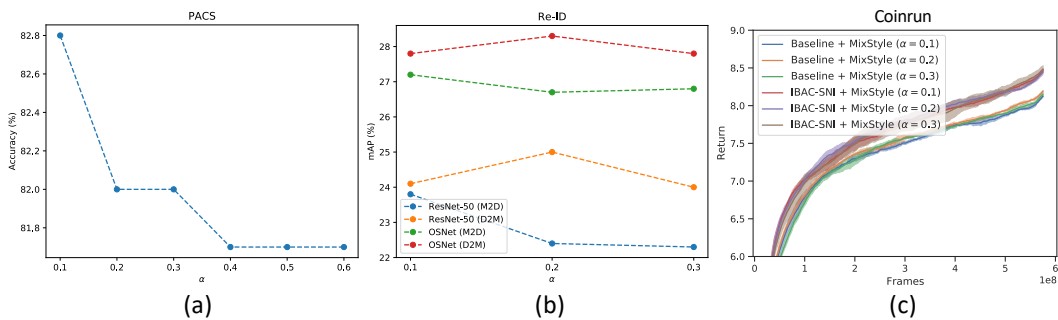

Figure 5: Evaluation on the hyper-parameter $\alpha$ on (a) PACS, (b) person re-ID datasets and (c) Coinrun. In (b), M and D denote Market1501 and Duke respectively.

**Mixing vs. replacing.** Unlike the AdaIN formulation, which completely replaces one style with the other, MixStyle mixes two styles via a convex combination. Table 4 shows that mixing is better than replacing. This is easy to understand: mixing diversifies the styles (imagine an interpolation between two data points).

Table 4: Mixing vs. replacing.

|  | Accuracy (%) |
|---|---|
| Mixing | **82.8**±0.4 |
| Replacing | 82.1±0.5 |

**Random vs. fixed shuffle at multiple layers.** Applying MixStyle to multiple layers, which has been shown advantageous in Table 3, raises another question of whether to shuffle the mini-batch at different layers or use the same shuffled order for all layers. Table 5 suggests that using random shuffle at different layers gives a better performance, which may be attributed to the increased noise level that gives a better regularization effect.

Table 5: Random vs. fixed shuffle at multiple layers.

|  | Accuracy (%) |
|---|---|
| Random | **82.8**±0.4 |
| Fixed | 82.4±0.5 |

**Sensitivity of hyper-parameter.** Recall that $\alpha$ is used to control the shape of Beta distribution, which has a direct effect on how the convex weights $\lambda$ are sampled. The smaller $\alpha$ is, the more likely the value in $\lambda$ is close to the extreme value of 0 or 1. In other words, a smaller $\alpha$ favors the style statistics in Eqs. (5) & (6) to be dominated by one side. We first evaluate $\alpha$ on PACS. Fig. 5(a) shows that with $\alpha$ increasing from 0.1 to 0.4, the accuracy slides from 82.8% to 81.7%. However, further increasing $\alpha$ does not impact on the accuracy. Therefore, the results suggest that the performance is not too sensitive to $\alpha$; and selecting $\alpha$ from $\{0.1, 0.2, 0.3\}$ seems to be a good starting point. We further experiment with $\alpha \in \{0.1, 0.2, 0.3\}$ on the re-ID datasets and the Coinrun benchmark. Figs. 5(b) & (c) show that in general the variance for the results of different values is small. Therefore, we suggest practitioners to choose $\alpha$ from $\{0.1, 0.2, 0.3\}$, with $\alpha = 0.1$ being a good default setting.

For more analyses and discussions, please see Appendix A.2.

## 4 RELATED WORK

**Domain generalization,** or DG, studies out-of-distribution (OOD) generalization given only source data typically composed of multiple related but distinct domains. We refer readers to Zhou et al. (2021) for a comprehensive survey in this topic. Many DG methods are based on the idea of aligning features between different sources, with a hope that the model can be invariant to domain shift given unseen data. For instance, Li et al. (2018b) achieved distribution alignment in the hidden representation of an autoencoder using maximum mean discrepancy; Li et al. (2018c) resorted to adversarial learning with auxiliary domain classifiers to learn features that are domain-agnostic. Some works explored domain-specific parameterization, such as domain-specific weight matrices (Li et al., 2017) and domain-specific BN (Seo et al., 2020). Recently, meta-learning has drawn increasing attention from the DG community (Li et al., 2018a; Balaji et al., 2018; Dou et al., 2019). The main idea is to expose a model to domain shift during training by using pseudo-train and pseudo-test domains, both drawn from source domains. Data augmentation has also been investigated for learning domain-invariant models. Shankar et al. (2018) introduced a cross-gradient training method (Cross-Grad) where source data are augmented by adversarial gradients obtained from a domain classifier.

Gong et al. (2019) proposed DLOW (for the DA problem), which models intermediate domains between source and target via a domainness factor and learns an image translation model to generate intermediate-domain images. Very recently, Zhou et al. (2020a) introduced L2A-OT to learn a neural network to map source data to pseudo-novel domains by maximizing an optimal transport-based distance measure. Our MixStyle is related to DLOW and L2A-OT in its efforts to synthesizing novel domains. However, MixStyle differs in the fact that it is done implicitly with a much simpler formulation leveraging the feature-level style statistics and only few lines of extra code on top of a standard supervised classifier while being more effective. Essentially, MixStyle can be seen as *feature*-level augmentation, which is clearly different from the *image*-level augmentation-based DLOW and L2A-OT.

**Generalization in deep RL** has been a challenging problem where RL agents often overfit training environments, and as a result, perform poorly in unseen environments with different visual patterns or levels (Zhang et al., 2018a). A natural way to improve generalization, which has been shown effective in (Cobbe et al., 2019; Farebrother et al., 2018), is to use regularization, e.g., weight decay. However, Igl et al. (2019) suggested that stochastic regularization methods like dropout and batch normalization (which uses estimated population statistics) have adverse effect as the training data in RL are essentially model-dependent. As such, they proposed selective noise injection (SNI), which basically combines a stochastic regularization technique with its deterministic counterpart. They further integrated SNI with information bottleneck actor critic (IBAC-SNI) to reduce the variance in gradients. Curriculum learning has been investigated in (Justesen et al., 2018) where the level of training episodes progresses from easy to difficult over the course of training. Gamrian & Goldberg (2019) leveraged the advances in GAN-based image-to-image translation (Liu et al., 2017) to map target data to the source domain which the agent was trained on. Tobin et al. (2017) introduced domain randomization, which diversifies training data by rendering images with different visual effects via a programmable simulator. With a similar goal of data augmentation, Lee et al. (2020) pre-processed input images with a randomly initialized network. Very recent studies (Laskin et al., 2020; Kostrikov et al., 2020) have shown that it is useful to combine a diverse set of label-preserving transformations, such as rotation, shifting and Cutout. Different from the aforementioned methods, our MixStyle works at the feature level and is orthogonal to most existing methods. For instance, we have shown in Sec. 3.3 that MixStyle significantly improves upon IBAC-SNI.

## 5 CONCLUSION

We presented a simple yet effective domain generalization method, termed MixStyle. MixStyle mixes the feature statistics of two instances to synthesize novel domains, which is inspired by the observation in style transfer work that the feature statistics encode style/domain-related information. Extensive experiments covering a wide range of tasks were conducted to demonstrate that MixStyle yields new state-of-the-art on three different tasks.

ACKNOWLEDGMENTS

This work was supported in part by the Shanghai Committee of Science and Technology, China (Grant No. 20DZ1100800), in part by the National Natural Science Foundation of China under Grant (61876176, U1713208), the National Key Research and Development Program of China (No. 2020YFC2004800), and in part by the Guangzhou Research Program (201803010066).

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

# A  APPENDIX

## A.1  PSEUDO-CODE OF MIXSTYLE

Algorithm 1 provides a PyTorch-like pseudo-code.

---

**Algorithm 1** PyTorch-like pseudo-code for MixStyle.

---

```
# x: input features of shape (B, C, H, W)
# p: probabillity to apply MixStyle (default: 0.5)
# alpha: hyper-parameter for the Beta distribution (default: 0.1)
# eps: a small value added before square root for numerical stability (default: 1e-6)

if not in training mode:
    return x

if random probability > p:
    return x

B = x.size(0) # batch size

mu = x.mean(dim=[2, 3], keepdim=True) # compute instance mean
var = x.var(dim=[2, 3], keepdim=True) # compute instance variance
sig = (var + eps).sqrt() # compute instance standard deviation
mu, sig = mu.detach(), sig.detach() # block gradients
x_normed = (x - mu) / sig # normalize input

lmda = Beta(alpha, alpha).sample((B, 1, 1, 1)) # sample instance-wise convex weights

if domain label is given:
    # in this case, input x = [x^i, x^j]
    perm = torch.arange(B-1, -1, -1) # inverse index
    perm_j, perm_i = perm.chunk(2) # separate indices
    perm_j = perm_j[torch.randperm(B // 2)] # shuffling
    perm_i = perm_i[torch.randperm(B // 2)] # shuffling
    perm = torch.cat([perm_j, perm_i], 0) # concatenation
else:
    perm = torch.randperm(B) # generate shuffling indices

mu2, sig2 = mu[perm], sig[perm] # shuffling
mu_mix = mu * lmda + mu2 * (1 - lmda) # generate mixed mean
sig_mix = sig * lmda + sig2 * (1 - lmda) # generate mixed standard deviation

return x_normed * sig_mix + mu_mix # denormalize input using the mixed statistics
```

---

Table 7: Test results on the source domains on PACS. A: Art. C: Cartoon. P: Photo. S: Sketch.

| Method | C,P,S | A,P,S | A,C,S | A,C,P | Avg |
|---|---|---|---|---|---|
| Vanilla | 99.49±0.03 | 99.47±0.04 | 99.38±0.02 | 99.65±0.03 | 99.50 |
| MixStyle | **99.55**±0.02 | **99.54**±0.01 | **99.47**±0.03 | **99.68**±0.03 | **99.56** |

Table 8: Leave-one-domain-out generalization results on Digits-DG.

| Method | MNIST | MNIST-M | SVHN | SYN | Avg |
|---|---|---|---|---|---|
| JiGen | 96.5 | 61.4 | 63.7 | 74.0 | 73.9 |
| CCSA | 95.2 | 58.2 | 65.5 | 79.1 | 74.5 |
| MMD-AAE | 96.5 | 58.4 | 65.0 | 78.4 | 74.6 |
| CrossGrad | **96.7** | 61.1 | 65.3 | 80.2 | 75.8 |
| L2A-OT | **96.7** | **63.9** | **68.6** | **83.2** | **78.1** |
| CNN | 95.8±0.3 | 58.8±0.5 | 61.7±0.5 | 78.6±0.6 | 73.7 |
| + Mixup w/o label interpolation | 93.7±0.6 | 55.2±1.0 | 61.6±0.9 | 74.4±0.8 | 71.2 |
| + Manifold Mixup | 92.7±0.4 | 53.1±0.8 | 64.4±0.2 | 76.8±0.5 | 71.7 |
| + CutMix | 94.9±0.2 | 50.1±0.5 | 64.1±0.9 | 78.1±0.7 | 71.8 |
| + Mixup | 94.2±0.5 | 56.5±0.8 | 63.3±0.7 | 76.7±0.6 | 72.7 |
| + Cutout | 95.8±0.4 | 58.4±0.6 | 61.9±0.9 | 80.6±0.5 | 74.1 |
| + DropBlock | 96.2±0.1 | 60.5±0.6 | 64.1±0.8 | 80.2±0.6 | 75.3 |
| + MixStyle (ours) | 96.5±0.3 | 63.5±0.8 | 64.7±0.7 | 81.2±0.8 | 76.5 |

## A.2 FURTHER ANALYSIS

**MixStyle between same-domain instances.**
We are interested in knowing if mixing styles between same-domain instances helps performance. To this end, we sample each mini-batch from a single domain during training when using MixStyle. The results are shown in Table 6 where we observe that mixing styles between same-domain instances is about 1% better than the baseline model. This suggests that instance-specific style exists. Nonetheless, the performance is clearly worse than mixing styles between instances of different domains.

Table 6: Investigation on the effect of mixing styles between same-domain instances.

|  | Accuracy (%) |
|---|---|
| ResNet18 | 79.5 |
| + MixStyle w/ same-domain | 80.4 |
| + MixStyle w/ random shuffle | 82.8 |
| + MixStyle w/ domain label | **83.7** |

**Performance on source domains.** To prove that MixStyle does not sacrifice the performance on seen domains in exchange for gains on unseen domains, we report the test accuracy on the held-out validation set of the source domains on PACS in Table 7.

**Results on Digits-DG and Office-Home.** In addition to the experiments on PACS (in Sec. 3.1), we further evaluate MixStyle's effectiveness on two DG datasets, namely Digits-DG (Zhou et al., 2020a) and Office-Home (Venkateswara et al., 2017). Digits-DG contains four digit datasets (domains) including MNIST (LeCun et al., 1998), MNIST-M (Ganin & Lempitsky, 2015), SVHN (Netzer et al., 2011) and SYN (Ganin & Lempitsky, 2015). Images from different digit datasets differ drastically in font style, stroke color and background. Office-Home is composed of four domains (Artistic, Clipart, Product and Real World) with around 15,500 images of 65 classes for home and office object recognition. The results are shown in Tables 8 and 9 where no domain labels are used in MixStyle. Similar to the results on PACS, here we observe that MixStyle also brings clear improvements to the baseline CNN model and outperforms all general-purpose regularization methods on both Digits-DG and Office-Home. Compared with more sophisticated DG methods like L2A-OT, MixStyle's performance is comparable, despite being much simpler to train and consuming much less computing resources.

Table 9: Leave-one-domain-out generalization results on Office-Home.

| Method | Artistic | Clipart | Product | Real World | Avg |
|---|---|---|---|---|---|
| JiGen | 53.0 | 47.5 | 71.5 | 72.8 | 61.2 |
| CCSA | 59.9 | 49.9 | 74.1 | 75.7 | 64.9 |
| MMD-AAE | 56.5 | 47.3 | 72.1 | 74.8 | 62.7 |
| CrossGrad | 58.4 | 49.4 | 73.9 | 75.8 | 64.4 |
| L2A-OT | **60.6** | 50.1 | **74.8** | **77.0** | **65.6** |
| ResNet18 | 58.9±0.3 | 49.4±0.1 | 74.3±0.1 | 76.2±0.2 | 64.7 |
| + Manifold Mixup | 56.2±0.4 | 46.3±0.3 | 73.6±0.1 | 75.2±0.2 | 62.8 |
| + Mixup w/o label interpolation | 57.0±0.2 | 48.7±0.2 | 71.4±0.6 | 74.5±0.4 | 62.9 |
| + Cutout | 57.8±0.2 | 48.1±0.3 | 73.9±0.2 | 75.8±0.3 | 63.9 |
| + CutMix | 57.9±0.1 | 48.3±0.3 | 74.5±0.1 | 75.6±0.4 | 64.1 |
| + DropBlock | 58.0±0.1 | 48.1±0.1 | 74.3±0.3 | 75.9±0.4 | 64.1 |
| + Mixup | 58.2±0.1 | 49.3±0.2 | 74.7±0.1 | 76.1±0.1 | 64.6 |
| + MixStyle (ours) | 58.7±0.3 | **53.4**±0.2 | 74.2±0.1 | 75.9±0.1 | 65.5 |

