# OpenReview forum: "Domain Generalization with MixStyle"
_ICLR.cc/2021/Conference — ICLR 2021 Poster_

### Official Review · AnonReviewer4 · 2020-10-26
**Simple but effective contribution to domain generalization**

**Rating:** 7
**Confidence:** 4

**Review:**

**Summary:** The paper proposes a simple method for domain generalization where multiple source domains are given for a certain task (like image classification) and testing happens on an unseen domain. The authors are inspired by normalization-based style-transfer techniques (Adaptive InstanceNorm) and propose to mix the styles of different source domains to effectively increase diversity of domains during training.

**Pros:**

- Overall, this is a well written paper with a clear idea that is simple but intuitive.
- The idea is well described, put into context of prior work and empirically validated to improve results over various baselines.
- It is good to see experiments outside of plain image classification to validate the proposed idea.
- The analysis where to apply MixStyle is good and makes intuitive sense.


**Cons:**

- The relation to MixUp needs to be explained in more details. While related to the proposed MixStyle, MixUp creates a convex combination of both input and output spaces. I can believe that MixUp as a standard data augmentation gives worse results than a vanilla CNN (Table 1) but I would not fully agree with the statement "... which demonstrates the advantage of mixing style statistics at the feature level over mixing images at the pixel level" from page 4. MixUp also interpolates the output label space, so the advantage cannot be only attributed the placement of the mixing within the network instead of at the pixel level.
- As an additional baseline, one could use MixUp with a sampled lambda that is larger than 0.5 in all cases (like in [FixMatch. Sohn et al. NeurIPS'20]) but keeping the label from sample $x$ rather than interpolating with $\hat{x}$.

- I do not understand why the suffix "_x" is added to the analysis in Table 3. Is MixStyle applied after each convolutional layer or after each block in a ResNet architecture? Specifically, for "conv234_x", how often is the MixStyle layer added? (3 times or 3 * num_convs_in_block times?)

- For the ReID experiments, I think it should be better highlighted that the cross-dataset setup is the key difference to evaluations in prior work. This somehow gets almost unnoticed because the default setting of ReID is already considered a valid domain generalization task due to the new label space and camera views. This left me a bit confused about how RandomErase can be a widely used data augmentation technique for ReID when it gives worse results in the experiments from Table 2. This became clear to me only after reading the discussion in the last paragraph of Section 3.2.

- I would not make the statement that "... mixing is CLEARLY better than replacing" on page 7 (see Table 4) while also stating that "... with alpha increasing from 0.1 to 0.4, the accuracy SLIGHTLY slides from 82.8% to 81.7%". That "slight" change is larger than the "clear" gap before.


**Other notes and open questions:**
- MixUp was used successfully as regularization for semi-supervised learning (SSL) [MixMatch. Berthelot et al. NIPS'19]. Can MixStyle also be used for SSL?

---

> ### Author Response · Authors · 2020-11-18
> **Response to Reviewer #4**
>
> **Q1. Relation to Mixup and an additional baseline.**
>
> Thanks. We have now reworded some of the claims to clarify the relation to Mixup. We have added the suggested baseline to Table 1 (i.e. Mixup w/o label interpolation, following Sohn et al. NeurIPS’20) and updated the discussion in the “comparison with general-purpose regularization methods” part (in Sec.3.1).
>
> We also include the main results below. Mixup w/o label interpolation is 1% worse than Mixup. The results seem to support our claim that mixing style statistics at the feature level via MixStyle is more useful for the DG problem.
>
> Model | Art | Cartoon | Photo | Sketch | Avg
> :--- | :---: | :---: | :---: | :---: | :---:
> ResNet18 | 77.0 | 75.9 | 96.0 | 69.2 | 79.5
> Mixup w/o label interpolation | 74.7 | 72.3 | 93.0 | 69.2 | 77.3
> Mixup | 76.8 | 74.9 | 95.8 | 66.6 | 78.5
> MixStyle w/ random shuffle | 82.3 | 79.0 | 96.3 | 73.8 | 82.8
> MixStyle w/ domain label | 84.1 | 78.8 | 96.1 | 75.9 | 83.7
>
> **Q2. Notation in Table 3.**
>
> Sorry for confusion. MixStyle was applied after a residual block rather than every convolution layer in that block. We have improved the notation in Table 3 by changing “conv2/3/4/5_x” to “res1/2/3/4” for clarity. “res123” (originally “conv234_x”) means MixStyle is applied after the 1st, 2nd and 3rd residual blocks.
>
> **Q4. Clarification on the setting for the re-ID experiments.**
>
> Thanks for the suggestion. We have updated the text in Sec.3.2 (in the “dataset and implementation details” part) and the caption in Table 2 to highlight the cross-dataset setting.
>
> **Q5. Use of words.**
>
> Thanks. We have removed “clearly” and “slightly” in those sentences to avoid overstatement.
>
> **Q6. MixStyle on SSL.**
>
> Interesting suggestion. MixStyle was specifically designed for the DG problem, but it may have potential in solving other problems as well. The experiments on the SSL task are currently beyond the scope of this paper. We will leave this investigation for future work.

---

### Official Review · AnonReviewer1 · 2020-10-28
**Insufficient technical novelty and experimental validation**

**Rating:** 6
**Confidence:** 4

**Review:**

This work proposes a technique for domain generalization by mixing style of images from different domains. This work adopts a mix up style approach [A] for domain generalization. Different from [A], the paper proposes to conduct mix-up in the intermediate layers, in particular, instance normalization layers. The proposed approach diversifies the data implicitly and the experimental results show that the mix-style can improve domain generalization.

Overall the paper is well-written with plenty of details. I also appreciate the experimental analysis in Sec 3.4 and the variance reported in Table 1. However, I have several concerns regarding the paper:
- The technical novelty seems rather incremental. This method is an extension of [A] to the instance normalization layer. Similar strategies have been discussed in other works such [B] and [C]. However, these works are not discussed in terms of main similarities/differences.
- I also found the experimental validation not fully sufficient to grant publication. Currently the validation is only conducted on PACS, the improvement also seems limited. I believe validation on more datasets(such as Digits, Office-Home as used in L2A-OT) can further confirm the effectiveness of the proposed method.
- I suspect that  interpolating the style parameter might cause performance drop on the domains that have been seen during training. Would it be possible to report performance on the domains that have been seen in the training?

[A] Vikas Verma et al. Manifold Mixup: Better Representations by Interpolating Hidden States. In ICML 2019.
[B] Rui Gong et al. DLOW: Domain Flow for Adaptation and Generalization. In CVPR 2019.
[C] Seonguk Seo. Learning to Optimize Domain Specific Normalization for Domain Generalization. In ECCV 2020.

---
I have read authors' response and other reviews. Some of my concerns are addressed in the response. Especially the added discussion with related work is helpful. Thus I would increase my rating to 6.

---

> ### Author Response · Authors · 2020-11-18
> **Response to Reviewer #1**
>
> **Q1. Novelty.**
>
> We would like to highlight that mixing style statistics at the feature level has never been introduced in the context of DG before, and we have demonstrated its effectiveness on a wide range of tasks in this paper. We also showed that MixStyle *largely* outperforms Mixup (Zhang et al. ICLR’18) and its extension ManifoldMixup (Verma et al. ICML’19) in the DG setting, as well as outperforming other general-purpose regularization methods.
>
> Compared with the two referenced works [B, C], MixStyle is novel. Specifically, [B] tackles domain adaptation by modeling intermediate domains between source and target via a domainness factor and learning a CycleGAN model for image translation. This is very different from mixing the style statistics at the feature level done in MixStyle, which is much simpler in implementation. Furthermore, compared with L2A-OT (Table 1), which shares a similar spirit with [B] in terms of generating novel images at the pixel level, MixStyle achieves highly comparable performance, despite consuming much less computing resources. Therefore, we think it is worth sharing our idea of MixStyle to the DG community and to encourage more research in the direction of exploiting feature-level style statistics. [C] learns domain-specific BN layers, so it has a much different motivation than that of MixStyle. We have now added the discussion on these two works in the related work section.
>
> **Q2. Experimental evaluation.**
>
> We would like to emphasize that our main goal is to evaluate MixStyle on a wide range of domain generalization tasks beyond the traditional image classification problem. Due to the space constraint, we were not able to present more extensive results on each of the three tasks. The results on the three different applications (image classification on PACS, instance retrieval on the re-ID datasets and RL on Coinrun) have demonstrated that MixStyle achieves significant improvements over the baselines. It is noteworthy that none of the existing domain generalization methods has demonstrated effectiveness on these datasets jointly.
>
> As requested, we have now conducted experiments on Digits and Office-Home, and have added the results to Table 8 and Table 9 respectively in the Appendix (A.2). The conclusion is similar to that on PACS: a) MixStyle brings clear improvements over the baseline CNN model; b) MixStyle outperforms all general-purpose regularization methods; c) MixStyle’s performance is comparable to more sophisticated DG methods such as L2A-OT, despite being much simpler to train and consuming much less computing resources.
>
> **Q3. Report performance on seen domains.**
>
> We have added these results in Table 7 in the Appendix (A.2). The results suggest that MixStyle does not sacrifice the performance on seen domains in exchange for gains on unseen domains. In fact, the source domains’ recognition accuracy is also improved very slightly. This is because for the source domain, MixStyle can be considered as a conventional data augmentation strategy. It thus is also able to improve the model’s generalization to test data, even when the domains are unchanged.

---

### Official Review · AnonReviewer3 · 2020-10-31
**Review for "domain generalization with MixStyle"**

**Rating:** 7
**Confidence:** 4

**Review:**

** Paper Summary **

This paper proposed a simple regularization technique for domain generalization tasks, termed MixStyle, based on the observation that domains are determined by image styles. By mixing styles of different instances, which generates synthesized domain samples while preserving the content features, the proposed method achieves the generalizability of the trained model. The MixStyle was applied to numerous applications, such as category classification, instance retrieval, and reinforcement learning, and attained the state-of-the arts. The MixStyle is relatively simple to implement, but effective.

** Paper Strength **
+ Simple methodological design, so it is easy to implement.
+ Understanding the domain shift problems as a style variation makes sense.
+ Randomizing the styles might be the solution to alleviate the domain generalization problems, but searching all the possible styles and applying them would be challenging and not feasible. So, using different instance samples to extract the styles was nice.
+ It makes sense that introducing the \lambda to mix the styles itself and ones of different instances.
+ The paper is well organized and written.

** Paper Weakness **

I have no major comments on this paper, but minor comments as follows:
- Even though the authors have shown the ablation study to analyze the levels where the MixStyle should be applied, it is not clear for me yet. The authors applied the MixStyle after 1st, 2nd, and 3rd residual blocks for category classification problems, but applied the MixStyle after 1st and 2nd residual blocks for category classification problems for instance retrieval task. In 3.4 analysis, they only showed the ablation studies on the category classification. Thus, one think the optimal combinations may vary according to the applications. In addition, another combination, e.g., conv34, conv25, would be more interesting.
- Fig 4 is hard to understand; what do the corresponding style statistics mean? Why does (d) only represent different legends?
- In Table 1, some experimental settings, e.g., Cartoon or Photo, have shown that MixStyle w/ random shuffle was better? The discussion on this might be interesting.

---

> ### Author Response · Authors · 2020-11-18
> **Response to Reviewer #3**
>
> **Q1. More analysis on where to apply MixStyle**
>
> Thanks. We have now followed the suggestion and conducted the ablation study on the re-ID datasets in Table 3.  We also included more combinations as suggested. The notation has been improved (using “res1” to denote the 1st residual block, “res12” to denote both the 1st and 2nd residual blocks, and so forth). Indeed, where best to add MixStyle is application dependent to some extent, but the conclusion is similar: 2-3 blocks at the very bottom are the best place to add MixStyle and anywhere near the top (classification layer) must be avoided. Please see the updated discussion in the “Where to apply MixStyle” part (in Sec.3.4) for details.
>
> **Q2. Clarification on Fig.4.**
>
> Each column corresponds to a residual block (from res1 to res4). The top shows the t-sne plot of feature maps, say from res1, while the bottom shows the t-sne plot of res1’s style statistics (i.e. concatenation of mean and std). We use these two plots together to show that low- and mid-level features encode style information while high-level features focus more on semantics, and therefore, MixStyle should be applied to the low- and mid-level features.
>
> The legend in Fig.4(d) is different from those in (a-c) because we want to show that, when we reach the top of the CNN (res4), the domain information is gone, and the clusters in (d) are related to class rather than domain. So adding MixStyle at res4 is not meaningful.
>
> **Q3. Discussion on why MixStyle w/ random shuffle is better on some domains.**
>
> Good point. The reason might be because there exist sub-domains in a source domain. For example in Fig. 1 and  Fig.4(a-c), it is clear that in each domain, there are distinct clusters in the style statistics distribution which correspond to sub-domains.  So these sub-domains, random shuffling could produce more diverse “new” domains that lead to a more domain-generalizable model. This discussion has been added to Sec.3.1 in the last sentence in the “comparison with general-purpose” part.

---

### Comment · ~Kaiyang_Zhou1 · 2021-03-07
**A new survey on domain generalization**

For readers who are interested in the topic of domain generalization (DG), we would like to share our recently released survey on DG at https://arxiv.org/abs/2103.02503.

**Why writing this survey?**

DG has undergone a decade progress since its first introduction in 2011, leading to a plethora of methodologies developed from different angles, as well as covering various applications like object recognition, medical image segmentation, person re-identification, action recognition, and so on. DG is of great importance to practical applications where out-of-distribution (OOD) generalization is key to successfully deploying machine learning systems.

However, there is no such a survey paper to provide the research community with a clear picture on how DG has developed so far. We think it is time to write a paper to summarize the ten-year development in DG.

**What you can learn from this survey?**

This survey basically answers the following questions:
- What is the definition of DG?
- Why do we study DG (*the motivation*)?
- How does DG compare to related problems such as domain adaptation and transfer learning?
- How is a DG method typically evaluated?
- What are the common datasets used for benchmarking DG methods?
- What methodologies have been developed to tackle DG? Can we categorize them?
- What are missing in the current DG research? And how can we possibly address them to further this field?

---

### Decision · Program_Chairs · 2021-01-07
**Final Decision**

**Decision:**

Accept (Poster)

**Comment:**

All three reviewers recommend acceptance after the rebuttal stage, and the AC found no reason to disagree with them. The proposed method is simple and effective, and the concerns raised about experimental validation and novelty seem well addressed in the rebuttal.